# Nurses’ Attitudes and Perceptions Regarding Suicidal Patients: A Quasi-Experimental Study of Depression Management Training

**DOI:** 10.3390/healthcare12030284

**Published:** 2024-01-23

**Authors:** Yael Sela, Yossi Levi-Belz

**Affiliations:** Ruppin Academic Center, Emek Hefer 4025000, Israel; yossil@ruppin.ac.il

**Keywords:** suicide, mental health, community nurses, depression, vignette

## Abstract

Suicide prevention is a priority globally. Community nurses are on the frontline of healthcare, and thus well placed to identify those at risk of suicide and act to prevent it. However, they are often ill-equipped to do so. This study examines whether depression management training for nurses may also help them manage suicide-risk patients. Method: This quasi-experimental study used a questionnaire that included a randomly assigned textual case vignette, measures related to patient descriptions portrayed in the vignette, and demographic and clinical/training information. The participants were 139 Israeli nurses who were mostly Jewish, Israeli-born, and married women working as community nurses. Almost half had completed depression management training (DMT) in their routine work. Results: Nurses who completed depression management training were more likely than non-trainees to query the patient regarding mental status and suicide plans and were more likely to refer them to appropriate further treatment. The graduate nurses also reported higher self-competence and more positive attitudes regarding their ability to assess depression and suicide risk than nurses who had not received depression management training. Discussion: The results highlight the importance of depression management training, as suicide assessment and referral are among the major steps to suicide prevention.

## 1. Introduction

Every 40 s, someone dies by suicide. Globally, this means that there are approximately 800,000 suicides annually, with over 20 million suicide attempts [1]. These alarming numbers, together with the knowledge that suicide can be prevented [2], increase the vital importance of designing interventions to diminish suicide rates. One of the most important facilitators of suicide is the presence of mental disorders [3]. Approximately 90% of all suicides occur among individuals with active mental health diagnoses, clinical depression being the most prevalent [4,5]. Thus, risk factors for depression, such as anhedonia, deep sadness, sleep difficulty, and loss of appetite [6], should be thoroughly examined and, if diagnosed, should facilitate acute intervention by any community suicide gatekeeper [7].

Healthcare professionals are at the forefront of mental healthcare and can be viewed as among the most important gatekeepers of suicide [8]. The importance of an appropriate response to suicidal risk by primary physicians and community nurses is reflected in previous research, showing that 45% of patients who died by suicide had contact with these primary care providers in the month prior to their death and 80% within the previous year [9,10].

In addition to primary physicians, community nurses often interact with patients over time and routinely conduct many interviews and examinations, including coordinating patient care, acting as case managers, and increasing health literacy [11]. In Israel, community nurses work with the family physician as a team. Together, they comprise the primary patient care [12,13] and can play a critical role in suicide prevention. Knowing the patient and the family allows the nurse to detect a suicide risk and provide appropriate intervention [14].

However, several studies have indicated that some community nurses may not have the specific knowledge or confidence needed to interact with suicidal patients [15,16]. Nurses may not feel competent to assess suicide ideation and risk among patients. They acknowledge several psychosocial barriers and challenges in assessing and managing individuals at risk of suicide [17,18,19]. Thus, it is plausible to suggest that nurses’ lack of knowledge and training in suicide risk management may negatively impact healthcare delivery and patient safety [20].

Several countries, including the United States and the United Kingdom, have implemented suicide prevention and identification training programs among nurses and nursing students [21], and many other countries have adopted similar programs with nurses and other primary care staff. The training programs designed for this aim may be brief (a few hours) or more extensive (weeks), but all seek to improve knowledge about suicidal behavior, perceptions regarding suicide prevention, reluctance to engage with the patient, and self-efficacy in their professional role with patients [7]. Some of these programs showed a significant improvement in post-training knowledge and accuracy in identifying warning signs, risk factors, and the protective factors of suicide [22,23], whereas other programs demonstrated no significant improvement [8,24,25,26,27].

### The Current Study

Suicide assessment and referral are acknowledged to be among the major steps in suicide prevention [28]. However, little is known about what community nurses in Israel know and perceive about suicide risk. Moreover, while most nurses in Israel do not attend suicide prevention programs, many do learn how to detect and intervene in cases of depression, as this is part of the formal nursing education curriculum. Thus, we ask whether nurses having received depression management training (DMT) would report greater ability and self-competence in suicide risk management than non-trainees. This is the first study of its kind to investigate the effect that DMT will have on nurses working in the community. These results could have a significant impact on future intervention programming and subsequent training among nurses working in Israel.

Our objective was to identify and analyze the attitudes and perceptions of Israeli community nurses on potential patient suicide risk. To this end, we initiated this quasi-experimental study in order to understand the effectiveness of DMT on nurse confidence when managing at-risk patients.

We posited the following hypotheses:

**H1.** 
*Nurses who had received DMT would be more willing to refer the hypothetical patient to a mental health professional and more willing to ask him about his mental health issues and suicide plans than nurses who had not received DMT beyond personal and professional characteristics (e.g., seniority).*


**H2.** 
*The effect of DMT will be moderated by the severity of the hypothetical patient’s mental condition: As the patient’s severity level rises in the vignette, DMT nurse trainees would be more willing to refer the patient for treatment and would be more willing to ask the patient about his mental health issues and suicide plans; no such differences would be found among nurses who had not participated in DMT.*


**H3.** 
*Among the nurses, the DMT graduates would report more positive attitudes and greater self-competence in their ability to manage patients’ depression and suicide risk than non-trainees.*


## 2. Methods

### 2.1. Design

In this 2 × 3 quasi-experimental study, we used two independent variables, DMT training (yes/no) and suicide severity levels, which were manipulated with three conditions (depression, suicide ideation and suicidal behavior). Each participant was randomly assigned to one of the three experimental conditions of the suicide severity variable and was asked about his/her DMT training. The number of participants in each of the three suicide severity conditions ranged from 43 to 50; 90 participants did not complete the DMT training while 54 did complete it.

### 2.2. Procedure

After ethics approval was granted by [MASKED], the research questionnaire was distributed both digitally and manually among designated community health centers. Additionally, a link to an electronic copy of the questionnaire was distributed in closed social media groups whose sole subscribers were community nurses. Moreover, the authors forwarded email invitations that included the link to the online questionnaire to several mailing lists of community nurses in different institutions, requesting participation. Consistent with the snowball method of participant recruitment, all recipients were asked to forward the invitation to other community nurses. Within 4 weeks of the initial invitations, 139 nurses responded to the online questionnaire. The questionnaire included a short case study followed by several questions. Informed consent was obtained after explaining the research procedure.

In the study, community nurses read a vignette about a hypothetical patient who had suffered from a psychological crisis. The vignette continued with one of three severity levels regarding the patient’s suicide risk: a description of a depressive episode, a description of suicide ideation, or a description of a previous suicide attempt. Following the reading, the nurses were asked about their willingness to engage in three possible actions concerning the hypothetical patient: refer him to a mental health professional, ask him about his mental health issues (e.g., depression symptoms), and ask him about his suicide plans. Other studies have already used such quasi-experimental designs in order to examine differences in suicide risk assessment following vignette evaluations [27,28,29], which enable us to understand the differences in assessment in regard to changes in the patient descriptions. Following the reading of the vignette, the participants were asked to fill out the study questionnaire, including demographics. In this section, we asked the nurses about their attitudes and confidence in managing suicide risk. Importantly, we also asked nurses if they received DMT for nurses and used this question to split the participants into two groups: with/without DMT.

### 2.3. Measures

The study questionnaire included a randomly assigned case vignette, associated measures relating to character descriptions within the vignette, and sociodemographic and background questionnaires.

#### 2.3.1. The Vignettes

To manipulate the severity of suicide risk and assign the participants to the study’s conditions, study participants were randomly presented with one of three vignettes that matched the three study conditions. All texts began with the same background story and description; then, three increasingly severe levels of the patient’s suicidal risk were randomly assigned to the participants. The story described a hypothetical male patient experiencing health problems following a serious injury incurred while working as a security guard 2 months before. The vignette described his visit to the doctor for routine treatment of his injury. The narrative presented below was constructed based on previous research in the field [27,28,29]:


*Nadav is 45 years old, married with three small children, and working full-time as a hospital security guard. About two months ago, he fell and injured his right knee during work hours. Since then, he has been limited in his walking ability and suffers from continuous pain. Following the fall, he took a long sick leave and stayed at home. In the last two weeks, he has returned to part-time work. Nadav is scheduled to undergo orthopedic surgery on his injured knee in about a month, and he is coming to your clinic for an ECG before the surgery. Nadav has a BMI of 30 and unbalanced diabetes. He is aware of his excess weight and diabetes, but he does not adhere to a healthy diet or exercise due to his working late shifts. His blood tests reveal that in the last three years, Nadav has suffered from low hemoglobin levels and a lack of vitamin B12.*


Following the initial vignette, presented to all participants, the virtual hypothetical patient’s symptoms were presented according to the randomized narrative assigned to each participant. Under the first severity level, that of a hypothetical depressed patient (n = 41), depression symptoms taken from the Beck Depression Inventory [30] were adapted to construct a credible picture of a person suffering from a depressive episode: *… In the last few weeks, he has lost five kilograms and suffers from a loss of appetite. The pain makes it difficult for him to help at home, and he is also troubled by the family’s financial situation. He feels very sad and does not function normally. He says that he does not feel that his wife supports him enough and that they fight all the time. He is highly distressed and feels hopeless regarding the future.*

Under the second severity level, the hypothetical patient presented suicidal ideation (n = 51), depression symptoms, and suicidal thoughts. The suicide ideation description was derived from the Columbia Suicide Severity Rating Scale (C-SSRS) [31], a measure to diagnose suicide risk severity according to both behavior and presented symptoms. Specific C-SSRS items were selected to portray the realistic portrayal of a patient suffering from a crisis and presenting with suicidal ideation: *… In the last week, he cries a lot and wishes to go to sleep and not wake up. He has thoughts of killing himself and thinks about how he will do it…*

Under the third severity level, a hypothetical patient presented plans for suicide (n = 54), depression symptoms, and suicidal thoughts, including an actual suicide plan. The description of the suicide attempt was constructed based on the Columbia Suicide Severity Rating Scale [31] described above. Items of the C-SSRS were selected to construct a realistic case study, portraying the most accurate situation of a person suffering from a crisis and presenting actual plans for suicide: *… Three days ago, he started planning his suicide. He purchased a large number of painkillers; he planned to take them all when alone in his apartment…*

#### 2.3.2. Manipulation Check

Participants’ perception of suicide risk of the hypothetical patient was assessed with two questions (“To what extent do you think the patient is at risk for suicide?” and “To what extent do you think the patient could die by suicide in the near future?”). Responses were rated on a 10-point Likert-type scale (higher scores signifying higher risk). Because a positive significant Pearson correlation was found between the two items, (*r*(139) = 0.89, *p* < 0.001), both scores were combined into a single measure. Thus, in the data analysis, this measure served as an index to reflect the severity of the hypothetical patient’s suicide risk.

#### 2.3.3. Willingness to Refer to and Willingness to Inquire about Mental State and Suicide Risk

Scales for assessing the degree of willingness to refer the hypothetical patient for further psychological assessment and the degree of willingness to inquire about the patient’s mental state and his suicide plans were designed for this study. To formulate questions for examining these concerns, questions resembling those from previous studies [28,32,33,34,35] were adapted for the current study.

Operationally, three items were formulated for this scale, assessing:The degree of *willingness to refer* (“To what extent would you be willing to refer the patient to a social worker/psychologist/physician regarding his mental state?”);The degree of *willingness to ask the patient about his mental state* (“To what extent would you be willing to ask the patient about his mental state?”);The degree of *willingness to ask the patient about future suicide plans* (“To what extent would you be willing to ask the patient about his future suicide plans?”).

All items were rated on a 10-point Likert-type scale. We found moderate correlations between the three items, *r*_(146)_ = 0.45–0.57, allowing us to conclude that the items are similar but not overlapping.

#### 2.3.4. Attitudes and Self-Competence Regarding Depression and Suicide Risk Assessment

Scales for assessing the nurses’ attitudes and self-competence regarding depression assessment and suicide risk assessment were adapted from other studies and included in this study [28,36].

##### Attitudes Regarding Depression Assessment

Nurses were asked to rate their agreement with three items regarding the nurse’s role in depression assessment (e.g., “*Identifying signs of depression is a part of nurses’ role*”). The three items were combined into an index of attitudes regarding depression assessment, yielding good reliability for the current sample (Cronbach’s α = 0.92).

##### Attitudes Regarding Suicide Risk Assessment

Nurses were asked to rate their agreement with three items regarding the nurse’s role in suicide risk assessment (e.g., “*Identifying signs of suicide risk is part of nurses’ role*”). The three items were combined into an index of attitudes regarding suicide risk assessment, yielding good reliability for the current sample (Cronbach’s α = 0.95).

##### Self-Competence Regarding Depression Assessment

Nurses were asked to rate their competence to accurately assess a patient’s depression. Four items were formulated (e.g., “*I believe I have the competence to assess a patient’s depression*”), together comprising an index of self-competence regarding depression assessment, yielding good reliability for the current sample (Cronbach’s α = 0.90).

##### Self-Competence Regarding Suicide Risk Assessment

Nurses were asked to rate their ability to accurately assess a patient’s suicide risk (e.g., “*I believe that I have the competence to assess a patient’s suicide risk*”). These four items were combined into an index of self-competence regarding suicide risk assessment, yielding good reliability (Cronbach’s α = 0.94).

All items were rated on a 10-point Likert-type scale. We found moderate correlations, *r*_(146)_ = 0.58–0.69), allowing us to conclude that the items were similar but not overlapping.

#### 2.3.5. Demographics, Clinical Training and Practice Characteristics

Demographic and clinical practice and training information were collected. These included participants’ gender, age, years of professional seniority, and experience and training in DMT (of at least 4 h).

### 2.4. Data Analysis

The hypothetical patient’s suicidal risk severity, as the dependent variable, was manipulated, resulting in three severity level conditions: depression, suicidal ideation, and suicidal behavior. Each participant was randomly assigned to one of the three experimental conditions. The second independent variable was previous participation in DMT. The dependent variables were the nurses’ willingness to refer the patient to a mental health professional and to ask him about his mental state and suicide plans. In all analyses, age was used as a covariate.

Firstly, we described the sample in terms of demographic variables. Second, for a manipulation check, we conducted two ANOVA analyses in which the dependent variable was the perceived level of suicidal severity of the hypothetical patient. Then, a two-way MANCOVA (multivariate analysis of covariance) was performed to examine the effects of the hypothetical patient’s suicide risk severity and the nurses’ DMT on their willingness to refer the patient and ask about his mental state and suicide plans, with age as a covariate variable.

These were followed by two one-way MANCOVAs that examined the effect of DMT on nurses’ perceived competence and attitudes regarding suicide risk assessment and suicide prevention. Nurses’ age, years of nursing experience, and years of community nursing experience served as covariates. Analyses were performed using SPSS version 23.

## 3. Results

### 3.1. Description of the Sample

Among the participants, 123 identified as female, 104 were [MASKED]-born, 121 Jewish, and 130 worked as community nurses. The participants’ average age was 41.58, and their average time working as nurses was 15.33 years. Table 1 presents the participants’ demographics.

### 3.2. Manipulation Check

To determine whether the hypothetical patient descriptions adequately distinguished between the suicide severity conditions, we examined the difference in the rating of perceived levels of the hypothetical patient’s suicide risk. An ANOVA revealed a significant between-group difference, *F*_(2,142)_ = 72.14, *p* < 0.001, Eta^2^ = 47. Nurses receiving the suicide attempt condition (the highest severity) rated the hypothetical patient at the highest suicide risk, *M* = 8.56, *SD* = 1.77, followed by a relatively high rating in the suicide ideation condition (of moderately high severity), *M* = 7.28, *SD* = 2.28, and a relatively low suicide risk rating in the depression condition (the lowest examined severity level), *M* = 3.63, *SD* = 2.05. Moreover, we conducted a Pearson analysis in order to understand the relationship between age, years of professional seniority and DMT training. Age (r = −0.27) and seniority (r = −0.23) were both negatively correlated with DMT training.

### 3.3. The Effect of Suicide Severity and Nurses’ Training in Depression Management

To examine the effects of suicide severity and DMT graduates on the nurses’ anticipated willingness to refer the patient for further examination and to ask him about his mental status and suicide plans, a MACNOVA was conducted with age as a covariate variable. We found three significant group effects: for suicide severity, Wilks’ *F* approximation (6,272) = 10.63, *p* < 0.001, Eta^2^ = 0.09; for DMT, Wilks’ *F* approximation (3,136) = 4.69, *p* < 0.001, Eta^2^ = 0.19; and for the interaction between suicide severity and training, Wilks’ *F* approximation (6,272) = 2.85, *p* < 0.01, Eta^2^ = 0.06.

The results of the univariate ANCOVAs shown in Table 2 and Figure 1 and Figure 2 reveal that as the suicide risk severity increased, the nurses reported a greater willingness to refer the hypothetical patient to further examination, to ask about his mental status, and to ask him about his suicide plans. Moreover, significant differences were found regarding the nurses’ DMT: Nurses who had completed this training reported a greater willingness to ask the patient about his mental status and to ask him about his suicide plans. No significant interactions were found relating to these three outcome measures.

### 3.4. Nurses’ Attitudes toward Suicide Assessment as a Function of Their Training in Depression Management

In the study’s second phase, we sought to understand the nurses’ self-competence and attitudes regarding suicide risk assessment, suicide prevention, and the effect of DMT on these variables. To do so, we conducted a one-way MANCOVA, with DMT (yes/no) as the independent variable and nurses’ perceived competence and attitudes regarding suicide risk assessment and suicide prevention as dependent variables. Nurses’ age, years of nursing experience, and years of community nursing experience served as covariates.

Overall, a significant group effect was found for DMT. We found a significant group effect for suicide severity, Wilks’ *F* approximation (4,128) = 10.78, *p* < 0.001, Eta^2^ = 0.25. As seen in Figure 3, the separate ANCOVAs revealed that, compared with non-trainees, DMT graduates reported higher levels of self-competence regarding depression assessment, *F*_(1,131)_ = 40.19, *p* < 0.001, eta^2^ = 0.25, and suicide assessment, *F*_(1,131)_ = 18.39, *p* < 0.001, eta^2^ = 0.12. Furthermore, DMT graduates reported more positive attitudes than non-trainees regarding their abilities to assess depression, *F*_(1,131)_ = 23.04, *p* < 0.001, eta^2^ = 0.15, and suicide risk, *F*_(1,131)_ = 27.39, *p* < 0.001, eta^2^ = 0.17.

## 4. Discussion

This study’s primary aim was to shed light on the knowledge and perceptions of suicide risk among community nurses in Israel. More specifically, we sought to understand the extent to which nurses’ DMT could be effective in suicide risk management. To date, no study has examined these issues empirically.

We found that, as the severity of the suicide risk of the hypothetical male patient increased, the nurses overall reported a greater willingness to refer him for further examination and to ask him about his mental status and suicide plans. Importantly, we found a significant effect of DMT on the nurses’ willingness to manage suicide risk: DMT graduates reported a greater willingness to ask the patient about his mental status and suicide plans than nurses who had not completed a DMT program. Moreover, we found that DMT graduates reported greater self-competence in depression and suicide assessment and greater positive attitudes regarding their ability to assess depression and suicide risk.

These results align with several studies that have shown that training in depression assessment among community nurses is related to a greater ability to accurately identify depressive symptoms and a greater ability to refer patients for further evaluation [37]. Moreover, some studies have highlighted that a suicide training program can improve attitudes toward suicide prevention and depression among general practitioners [38,39]. Suicide training has also been shown to increase general practitioners’ confidence in addressing depression and suicide in their everyday practice [40]. Moreover, a brief suicide prevention educational program significantly improved attitudes, knowledge, and confidence regarding suicide assessment and intervention in nursing staff [41]. However, no study has examined how DMT can affect community nurses’ ability to identify suicide risk. Thus, the study findings contribute to the scientific literature, highlighting how training can enhance knowledge, perceptions, and professional confidence among community nurses, who serve on the frontlines of mental healthcare in the community.

Our findings reveal that nurses can identify suicide risk and distinguish between levels of suicide severity. However, it is critical to note differences in training background at the early suicide intervention phase: Nurses who completed DMT reported they would take more action to diminish suicide risk (e.g., ask the patient about suicide plans) than those who did not receive such training. These findings highlight the importance of DMT, as it empowers the nurses to take the next step, presenting the patients with questions about their emotional state and referring them for further treatment.

This challenge of appropriate intervention has been reported in studies where primary care physicians and nurses testify that even when they identify a mental factor that requires investigation, they have difficulty taking suitable action [18,42,43]. Several factors have been associated with these professionals’ reluctance to intervene: Scholars have noted that a key challenge was the fear that questions about suicide might cause patient nervousness and impair the therapeutic relationship [43]; nurses reported lacking confidence in their ability to debrief the patient and manage the suicide risk situation; moreover, nurses felt they lacked sufficient operational tools for managing a situation where patients require assistance and further referral [21]. Suicide prevention training relating to these issues has been shown to dramatically improve the prospect of nurses choosing to react and not disregard the situation [44]. In our study, however, we demonstrate that DMT can also comprise a critical step forward in this issue, given its contribution to nurses’ competence, and can provide nurses with the operational tools for intervention when suicide risk is identified.

The American Psychiatric Nurses Association confirmed that nurses play a critical role in depression and suicide prevention, and their competence in these areas can significantly impact patient outcomes [45]. Nurses need background knowledge relevant to their practice and the ability to apply it in different situations. Regarding depression and suicide prevention, nurses must have a solid understanding of the risk factors, warning signs, and evidence-based interventions for these conditions. This knowledge can help nurses identify patients at risk of suicide or those struggling with depression and then take the next step of referring these patients for further treatment [46].

Beyond knowledge and skills, studies have shown that nurses’ attitudes can have a significant impact on the quality of care and patient treatment. Research has shown that nurses and primary physicians who hold stigmatizing attitudes and negative beliefs toward mental illness and suicide may be less attuned to warning signs of depression or suicide risk among their patients, and fail to provide appropriate care, such as referral to mental health services or crisis intervention [15,47]. Furthermore, research has consistently demonstrated that healthcare providers tend to hold pessimistic views about the reality and the prospects of recovery, leading them to believe that “what they do doesn’t matter” [48]. Moreover, nurses holding stigmatizing attitudes may inadvertently impart these beliefs to their patients, leading to patients’ feeling shame, guilt, or hopelessness. These feelings can further exacerbate patients’ mental health symptoms and hinder their ability or desire to seek treatment [49]. Thus, it is plausible to suggest that nurses who have completed DMT are more willing to perform suicide risk assessment. This may be due to the training’s impact on reducing the stigma regarding suicide and increasing the nurses’ sense of competency, thus facilitating a greater likelihood of their intervention and offering assistance to at-risk patients.

## 5. Limitations

Several methodological limitations should be considered upon interpreting the study results. First, as the cases we presented were hypothetical, the results reflect what the nurses perceive as their potential actions and do not necessarily reflect their actual behavior upon encountering an actual patient in a real situation. Furthermore, the described vignette was presented clearly and explicitly, and the nurses had considerable time to decide how to proceed. Real-life clinic situations, however, are often much more nebulous, which could impede the inclination to intervene. Regarding ecological validity, the nurses participating in the study could focus on the case study without having to deal with waiting patients. The typical clinic workload and the need to care for many patients do not always allow the nurses to perform interventions efficiently. Further studies may enable us to examine the competence and willingness to perform suicide risk assessment in real time in an actual clinic. Additionally, evaluating participant ”willingness to take action” using only three questions is a limitation. Future studies should use more in-depth evaluation of planned behavior. A second limitation relates to the voluntary nature of the sample, which may be over-representative of nurses who feel more competent in suicide risk situations. A broader, more representative sample of community nurses from different settings would facilitate a fuller understanding of the impact of DMT training on suicide risk assessment in this population.

Third, an important limitation derives from the decision not to analyze or evaluate the content of the DMT or consider the time-elapsed program completion. Future research should evaluate the training program comprehensively, from initiation to one year follow-up. Factors to be examined should include programmatic content and participant’s change in attitude, diagnostic ability, and depression-related care delivery. Thus, closely monitoring the training would enable a systematic assessment of the program’s effects.

Notwithstanding these limitations, our results emphasize that routine and accessible training programs for suicide risk assessment and management for community nurses are one of the critical steps needed in this emerging field. Targeted suicide prevention training may still be the gold standard for identifying and intervening with at-risk patients. However, as depression is a major risk factor for suicide [50], depression-focused training for nurses can be an important prevention step as it increases nurses’ knowledge, behavioral skills, and confidence and may make a real contribution to suicide prevention. This step can help community nurses assume an important role in suicide prevention efforts globally and save at-risk patients’ lives.

## 6. Conclusions

Our results emphasize that suicide risk assessment and management training for community nurses are an important aspect of future professional continuing education. DMT graduates reported higher feelings of self-competence in depression and suicide assessment and higher positive attitudes regarding their ability to assess depression and suicide risk. Targeted suicide prevention training may still be the gold standard for identifying and intervening with at-risk patients. However, as depression is a major risk factor for suicide, depression-focused training for nurses can be an important prevention step as it increases nurses’ knowledge, behavioral skills, and confidence and may make a real contribution to suicide reduction. This step can help community nurses assume an important role in suicide prevention efforts globally and save at-risk patients’ lives.

## Figures and Tables

**Figure 1 healthcare-12-00284-f001:**
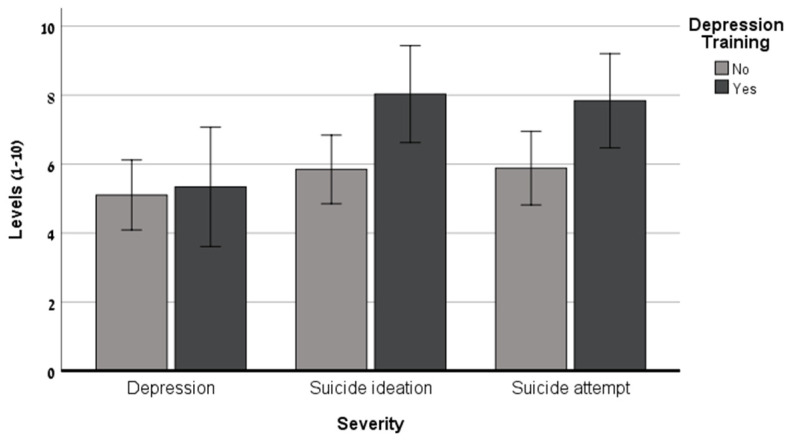
Willingness to ask about mental condition as a function of potential patient’ suicide severity and nurses’ depression training (N = 145).

**Figure 2 healthcare-12-00284-f002:**
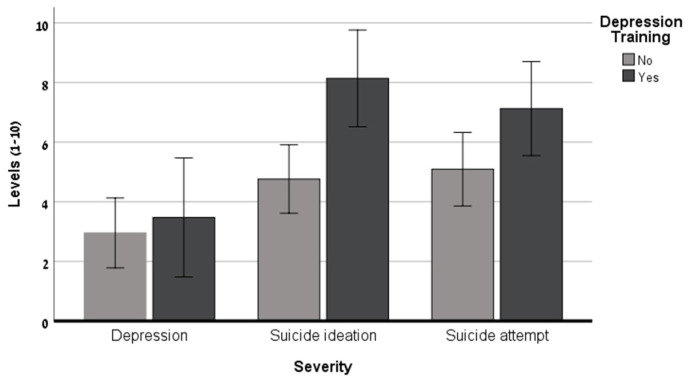
Willingness to ask about suicide plans as a function of potential patient’ suicide severity and nurses’ depression training (N = 145).

**Figure 3 healthcare-12-00284-f003:**
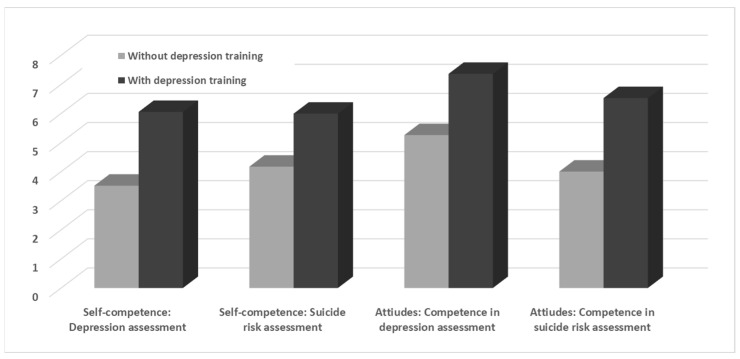
Nurses’ Self-Competence and Attitudes Regarding Depression and Suicide Risk Assessment as a Function of Depression Management Training (N = 145).

**Table 1 healthcare-12-00284-t001:** Participants’ Demographic and Professional Characteristics (N = 139).

Variables	N	%	Mean	*SD*	Range
Sex	139				
Male	16	11.5			
Female	123	88.5			
Birth country	139				
Masked-	104	74.8			
Masked-	35	25.2			
Religion	139				
Masked	121	87.1			
Masked	18	12.9			
Work	139				
Community care	130	93.5			
Other	9	6.5			
Age	138		41.58	10.46	24–62
Professional seniority	134		15.33	11.23	1–42

**Table 2 healthcare-12-00284-t002:** Willingness to Refer and Ask about Suicide as a Function of Suicide Severity of Hypothetical Patient Vignette and Training in Depression Management (N = 139).

		Suicide Severity Conditions			Statistical Analysis		
Outcome Measure	Training in Depression Management	Depression (n = 43) *M* (*SD*)	SI (n = 50) *M* (*SD*)	SA (n = 50) *M* (*SD*)	Main Effect—Severity	Main Effect—Training	Main Effect—Interaction
Willingness to refer to mental health professional	No (n = 90)	6.33	7.63	7.29	*F* = 6.65 *p* < 0.002 Eta^2^ = 0.09	*F* = 8.15 *p* < 0.005 Eta^2^ = 0.06	*F* = 1.36 *p* = 0.258 Eta^2^ = 0.02
		(2.84)	(2.67)	(2.51)			
	Yes (n = 54)	6.64	9.00	9.44			
		(2.84)	(2.67)	(2.51)			
Willingness to ask the patient about mental state	No (n = 90)	5.23	5.56	5.61	*F* = 3.21 *p* < 0.05 Eta^2^ = 0.04	*F* = 15.59 *p* < 0.001 Eta^2^ = 0.10	*F* = 1.74 *p* = 0.178 Eta^2^ = 0.02
		(2.87)	(2.95)	(2.67)			
	Yes (n = 54)	5.82	8.00	8.44			
		(2.92)	(2.81)	(2.33)			
Willingness to ask the patient about suicide plan	No (n = 90)	2.90	4.45	4.79	*F* = 11.77 *p* < 0.001 Eta^2^ = 0.15	*F* = 23.73 *p* < 0.001 Eta^2^ = 0.14	*F* = 2.34 *p* = 0.100 Eta^2^ = 0.03
		(2.59)	(3.56)	(3.40)			
	Yes (n = 54)	3.82	8.33	8.00			
		(2.89)	(2.94)	(3.28)			

## Data Availability

Data are contained within the article.

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
