# Peer review of "Nurses’ Attitudes and Perceptions Regarding Suicidal Patients: A Quasi-Experimental Study of Depression Management Training"

_healthcare, 2024, doi:10.3390/healthcare12030284_

Round 1

Reviewer 1 Report

Comments and Suggestions for Authors

Dear Authors, 

The issue is relevant and important, especially in the community and your research seems valuable from that point. 

I have some comments below. 

1. It would be nice if the research's purpose and hypothesis were described more clearly. I wondered below. 

DMT 's effectiveness has already been reported in previous studies. What is the novelty of this study? Why is your results meaningful beyond previous findings in other studies?  

2. I was aware that many contents of the paper could not be found in the cited references. And, some references are too out of date. It seems necessary to check the references again.

3. There are some editing errors in the manuscript, figure, and table.  

 I would suggest adding the number of each group in the table and editing the graphs. 

Comments on the Quality of English Language

I would suggest more editing of this paper.  

Reviewer 2 Report

Comments and Suggestions for Authors

Reviewer 3 Report

Comments and Suggestions for Authors

Title: It is advisable to include the study design in the title. Additionally, the title appears quite lengthy.

Abstract: The abstract seems appropriate.

Introduction: In line 37, you write (e.g., followed by a reference. I believe this might be a drafting error and needs reviewing.

The sentence from lines 74-77 is overly long, losing coherence in its reading. It requires revision.

Within lines 77-86 of the introduction, information is presented that seems more suitable for the methodology section, as it explains how certain variables were evaluated and how the sample was divided. Therefore, it might be more appropriate to remove this from the introduction. Perhaps it would be interesting to understand how these variables were previously evaluated to justify why this particular method of assessment and intervention was chosen over others. This information would be more fitting for the introduction.

Lastly, apart from stating the hypotheses, it is important to include the study's objective at the end of the introduction section.

Methods: The initial section should outline the study design. Additionally, it is important to specify the participant selection criteria. Were only community nurses considered?

Regarding sociodemographic characteristics, it would be suitable to specify which variables were measured, while the actual results should be included in the section bearing that title.

There are blinded elements mentioned in the text that aren’t blinded in the tables.

It isn’t clear how the suicide case was introduced. Was it presented when participants answered the sociodemographic and attitude questions, or was it a separate section within the questionnaire? This remains unclear.

Also, evaluating the participants' willingness to take action with only three questions seems insufficient.

Regarding the scale measuring attitudes towards depression and suicide risk, in line 196, several authors are cited in APA format, which would be incorrect. Moreover, the section ends with two colons, which is unclear whether it indicates missing text or if it's an error. Nevertheless, the adaptation from other studies is mentioned; was there a reliability and validity analysis conducted for this sample?

Sections 2.2.5., 2.2.6., 2.2.7., and 2.2.8. seem to be included within section 2.2.4., creating confusion in the organization of this part of the text.

It's unclear why section 2.2.9. introduces which sociodemographic variables were evaluated if these results were previously presented. Therefore, I reiterate that the sociodemographic characteristics table should be in the results section.

The data analysis section lacks a description of the descriptive analyses conducted.

Results: In line 271, is 'MANCOVA' intended?

Discussion: Well-structured and justified. The limitations are adequately highlighted, although they suggest that the study's contribution is quite limited.

Conclusions: The beginning of this section (line 389) might need revising. It is a separate section and should not refer to the limitations. Bibliographic references should not be included in the conclusions. This section needs improvement as it doesn't discuss the study's conclusions but rather the field of study.

Author Response

Please see th attachment
